# InfoBlend: Storing and Reusing KV Caches of Multimodal Information without Positional Restriction

## Abstract

The context caching technique is employed to accelerate the Multimodal Large Language Model (MLLM) inference by prevailing serving platforms currently. However, this approach merely reuses the Key-Value (KV) cache of the initial sequence of prompt, resulting in full KV cache recomputation even if the prefix differs slightly. This becomes particularly inefficient in the context of interleaved text and images, as well as multimodal retrieval-augmented generation. This paper proposes position-independent caching as a more effective approach for multimodal information management. We have designed and implemented a caching system, named InfoBlend, to address both system-level and algorithm-level challenges. InfoBlend stores the KV cache on local disks when receiving multimodal data, and calculates and loads the KV cache in parallel during inference. To mitigate accuracy degradation, we have incorporated the integrated reuse and recompute mechanism within the system. The experimental results demonstrate that InfoBlend can achieve up to 54% reduction in response time and $2\times$ improvement in throughput compared to existing context caching systems, while maintaining negligible or no accuracy loss.

## 1 Introduction

Recent years have witnessed notable development in applications based on Multimodal Large Language Model (MLLM), including those for code generation Zhu et al. (2023), graphical user interface agents Hong et al. (2024); Zhang et al. (2023a), and medical image understanding Li et al. (2023); Zhang et al. (2023b). To enhance the perceptual capabilities of MLLM, the number of processed image tokens has increased considerably, from 576 in LLaVA 1.5 Liu et al. (2024a) to 2304 in LLaVA 1.6 Liu et al. (2024b). This significant expansion in the number of tokens has the adverse effect of slowing down MLLM inference and constraining the performance of applications. This highlights the necessity for reducing latency and enhancing throughput.

In order to reduce the computational overhead, Context Caching (CC) is a prevalent technique employed by numerous prominent platforms, including Google Gemini Google (2024a), Moonshot Kimi Moonshot (2024), DeepSeek Deepseek (2024), vLLM Kwon et al. (2023), and SGLang Zheng et al. (2024). As users may access the same text or image context on multiple occasions, the intermediate results (commonly referred to as KV cache Pope et al. (2023)) of these information items can be stored for reuse. To illustrate, Gemini offers a CC API Google (2024b), which allows users to upload a video file in advance. Subsequently, the KV cache of the file and system prompt[1] is precomputed, and reused when responding to the user's query. In this manner, CC reduces the response time of MLLM, namely the Time-to-First-Token (TTFT).

However, all of the aforementioned CC systems merely reuse the KV cache of the initial sequence of tokens (the prefix). Given the autoregressive nature of MLLM, the KV value of each token depends on the preceding tokens. If the prefix of a query differs slightly from that of a previous query, the majority of existing CC systems will cease attempting to reuse the stored KV cache and instead recompute it for the new query. Such inefficiencies can prove particularly problematic in

---

[1]System prompt is a set of instructions or guidelines that inform the model about its role, the nature of the task, and the expected behavior.

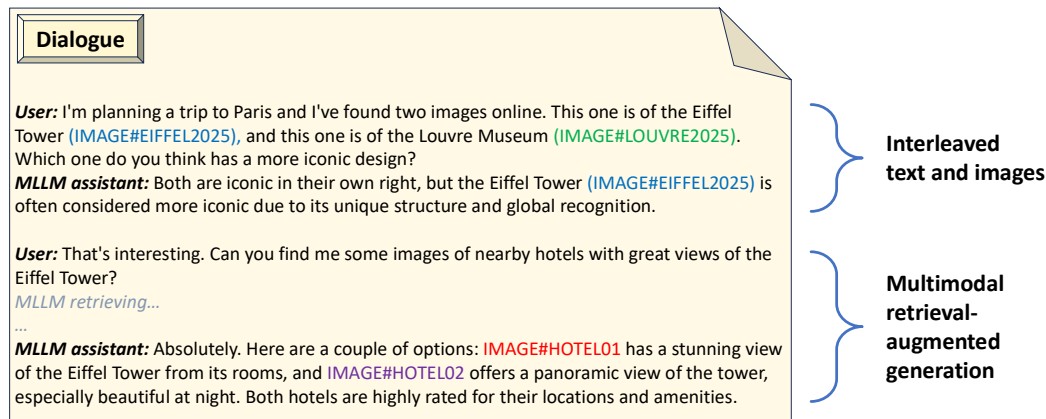

Figure 1: An example of a dialogue between a user and MLLM assistant. The first round of dialogue represents an example of interleaving text and images, whereas the second round of dialogue is an example of information retrieval. IMAGE#EIFEL2025 and IMAGE#LOUVRE2025 are two images uploaded by the user, while IMAGE#HOTEL01 and IMAGE#HOTEL02 are two images fetched from external websites by MLLM assistant. In both examples, the KV cache of images cannot be reused by prefix-based CC systems, as the opening words may differ.

cases where interleaved text and images Huang et al. (2024) are concerned, as well as in the context of Multimodal Retrieval-Augmented Generation (MRAG[2]) Wei et al. (2024). Suppose there is a dialogue as shown in Figure 1. If a subsequent query is the same as this one, except that it starts with "*We're planing to ...*", then the entire KV cache cannot be reused. The use of interleaved text and images is a widespread phenomenon on the Internet, particularly in the context of blogs and news media. MRAG fetches relevant information to meet specific user requirements, which is also an important function for applications.

In this paper, we explore the possibilities of position-independent CC systems. We start with analyzing the limitations of two existing approaches: prefix caching and full reuse. Experimental results show that full reuse can save up to 69.4% in TTFT, while concurrently leading to a substantial decline in generation quality (see details in § 2.2). Consequently, we propose partial reuse of KV cache as a trade-off between the two approaches.

The implementation of partial reuse faces both system-level and algorithm-level challenges. First, the size of a single image's KV cache can reach 1 GB, so the majority of KV caches are stored on the local disk. Second, it is crucial and challenging to avoid quality degradation when reusing KV cache. Third, recent studies on RAG and LLM serving systems Agarwal et al. (2025); Gim et al. (2024); Hu et al. (2025); Yao et al. (2025) undergo a two-step process in the field of MLLM. They assume that all chunks are concatenated consecutively, without considering interleaved variable text. In the context of interleaved text and images, they first compute the KV caches of text, and then reuse the KV caches of text and images, introducing additional time overhead during MLLM serving.

We address the challenges by designing a system named INFOBLEND. Analogous to classical position-independent code, INFOBLEND allows the KV caches to be reused at any position within a prompt, not merely at the prefix. To maintain generation quality, we have analyzed the attention mechanism of image tokens thoroughly, and select which tokens should be recomputed. Finally, we design and implement the selective attention mechanism to reuse the KV caches in single step.

Evaluations on multiple MLLMs and datasets illustrate that INFOBLEND saves TTFT by 54.1% compared to prefix caching, with accuracy loss within 13.6%. In the scenario of online services, IN-FOBLEND improves the throughput by 2.0× compared to the state-of-the-art approach. In addition, the generation quality of INFOBLEND does not degrade as the number of images grows.

---

[2]MRAG refers to a retriever that retrieves multimodal data.

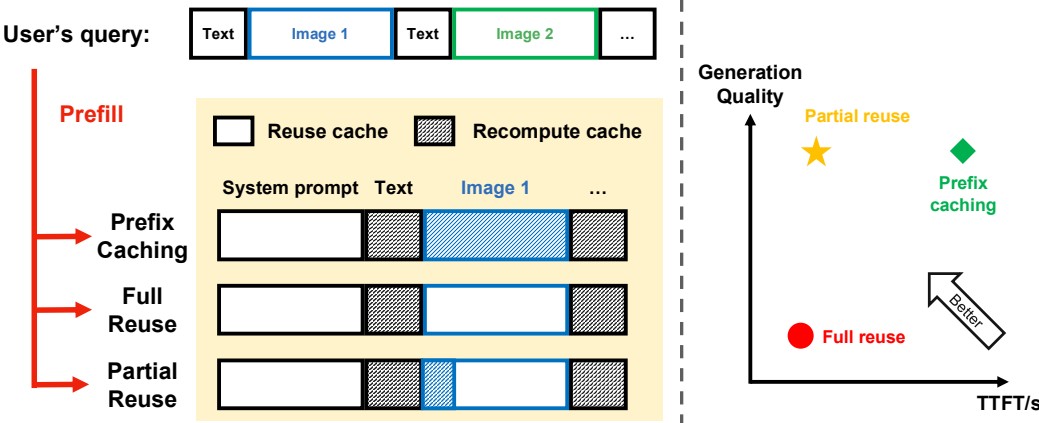

Figure 2: Illustration of three processing methods. **Left part**: A query consists of multiple images and parts of text. Suppose that the first sentence of the query does not match any other previous queries, and the KV caches of images are stored in advance. *Prefix caching* reuses the KV cache of system prompt, and recomputes that of the query. *Full reuse* recomputes the KV cache of text, and reuses that of multimodal information. *Partial reuse* recomputes more tokens compared to *Full reuse*. **Right part**: *Partial reuse* accelerates MLLM inference through reusing the KV caches of the most image tokens, while maintaining generation quality by selectively recomputing.

## 2 BACKGROUND AND MOTIVATION

### 2.1 BENEFITS OF CONTEXT CACHING (CC)

The CC technique has been shown to enhance the efficiency and performance of the MLLM through the storage and reuse of the context of prior interactions. In a video analysis task, for instance, the user is likely to refer to the video multiple times, and reusing the KV cache of the video can reduce the response time and enhance the user experience, especially when the video is of considerable length. Similar beneficial scenarios for CC include code analysis with repeated repository references, and Q&A assistants with a collection of pictures.

The adoption of CC techniques offers notable advantages for both service platforms and users. For service platforms, CC contributes to a reduction in computational overhead, enabling providers to accommodate a greater number of users. Consequently, MLLM service providers actively promote the use of CC by users. For example, DeepSeek cuts API costs by up to 90% for cache hits Deepseek (2024). This incentivizes users to employ the CC API in pursuit of reduced costs.

### 2.2 LIMITATIONS OF TWO NAIVE APPROACHES

At present, nearly all CC systems are of the prefix-based variety. In the remainder of this paper, the term "*prefix caching*" is employed to denote the prefix-based CC system, as it merely stores and reuses the KV cache of the prefix. We detail the prefix caching in Appendix A. This approach is constrained by the necessity of an exact match in the prefix tokens across requests. We illustrate this phenomenon in the left part of Figure 2. When the prefix of a new query differs from that of any other previous query, prefix caching recomputes all the tokens except the system prompt. This turns out to be inefficient, since the number of tokens of multimodal information is considerably greater than that of text. When the user's query becomes longer, prefix caching will be nearly as slow as computing without the CC system.

In order to explore the potential for reducing TTFT, we implemented a naive approach called "*full reuse*" which reuses the entire KV cache regardless of the position of multimodal data. Nevertheless, the text input of the user is not cached as it is unpredictable. As shown in the left part of Figure 2, full reuse first recomputes the KV cache of text, and then concatenates it with stored KV caches, and finally computes the first output token with all KV caches. This approach is analogous to Prompt Cache Gim et al. (2024). We conduct an experiment to compare prefix caching and full reuse, and

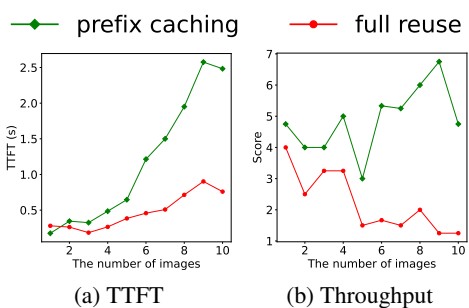

(a) TTFT    (b) Throughput

Figure 3: Comparison between prefix caching and full reuse in terms of TTFT and generation quality. The MLLM model is LLaVA-1.6-mistral-7B and the dataset is MMDU. (a) The TTFT of full reuse grows slowly. (b) The score in is assessed by ChatGPT, which is a common practice in evaluating open questions. Full reuse cannot match the performance of prefix caching, especially when the number of images is large.

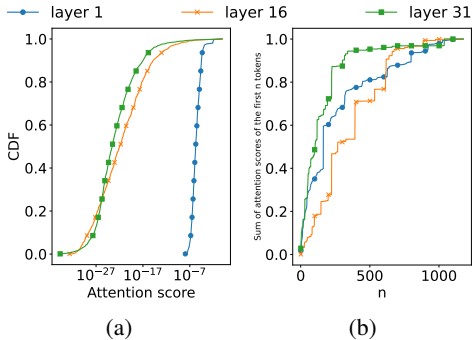

(a)    (b)

Figure 4: (a) Distribution of all image tokens in terms of computed attention score with the first output token. Note that the x-axis is expressed on a log scale. (b) The summation of attention scores of the first $n$ image tokens. For the sake of clarity, we show only three representative layers, and other layers show similar patterns.

the results are shown in Figure 3. The TTFT of prefix caching grows quadratically with respect to the number of images, since the computational complexity of the attention mechanism is $O(n^2)$, where $n$ is the length of the prompt. In contrast, the TTFT of full reuse grows slowly, due to the reuse of KV cache. When the number of image is large, full reuse can reduce the TTFT by 69.4% compared to prefix caching.

However, it is clear that full reuse violates the attention mechanism in transformer models. The stored KV cache differs from the required one due to the autoregressive nature of transformer. Figure 3b illustrates that full reuse degrades the generation quality significantly. As the number of images increases, this approach is incapable of producing any meaningful answers. Moreover, full reuse is a two-step process: The first step is to calculate the KV cache of text, and the second step is to calculate the first output token using the concatenated KV cache. This triggers the LLM engine twice, introducing additional time overhead from a system perspective. Figure 3a demonstrates this inefficiency. When the number of images is 1, the TTFT of full reuse is larger than that of prefix caching, due to the two-step process.

In summary, prefix caching is inefficient in decreasing TTFT, while full reuse exhibits poor performance in generation quality. In the rest of this paper, we aim to find a trade-off between prefix caching and full reuse.

### 2.3 OPPORTUNITIES OF PARTIAL REUSE

This section explores the possibility of improving generation quality through minor modifications to the stored KV caches. Recall that the KV cache of multimodal data is computed through modality encoder, connector, and self-attention. We posit that the KV cache contains the majority of the multimodal information, except for position information and cross-attention with other inputs. Our goal is to blend the missing position knowledge into the KV cache. This goal can be achieved through integrated reuse and recompute mechanism, named "*partial reuse*". As illustrated in Figure 2, partial reuse recomputes a few tokens to mitigating accuracy degradation.

In order to validate this idea, we conduct an experiment using the example in Figure 1. The utilized MLLM is LLaVA-1.6-vicuna-7B Liu et al. (2024b). It encodes the image "IMAGE#EIFFEL2025" into 1176 tokens. We collect the attention scores between these image tokens and the first output token from each transformer layer. Their values are normalized with softmax function. Figure 4a shows the cumulative distribution function (CDF) of these attention scores. We find that less than 5% of tokens receive more than $10^{-3}$ attention score. Since $10^{-3}$ is small, this means that less than 5% of tokens affect the output. Many recent papers also point out this phenomenon Beltagy et al. (2020); Chen et al. (2025); Tang et al. (2024); Zhang et al. (2024; 2025). We conclude it as Insight 1.

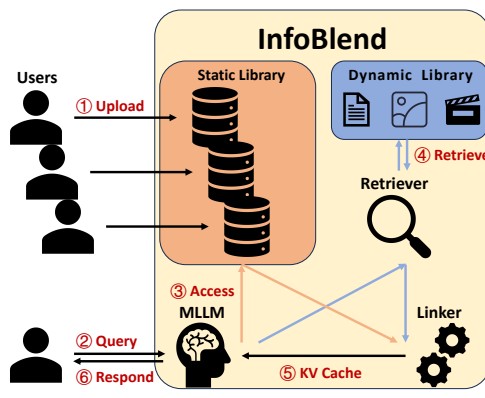

Figure 5: The Architecture of INFOBLEND Serving System.

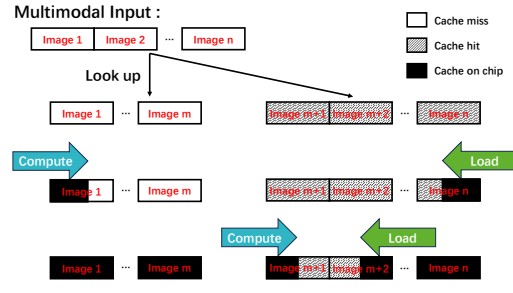

Figure 6: Load and compute the KV cache of images in parallel. Note that we simply utilize images as an example here. The parallelization is applicable to the other forms of multimodal data as well.

**Insight 1.** *Attention matrix is extremely sparse.*

To find out which tokens receive more attention, we calculate the cumulative summation of attention scores as shown in Figure 4b. It is evident that the first 500 tokens account for approximately 80% of attention scores. This tendency aligns with the human attention mechanism, in which individuals tend to allocate more attention to the initial sentences of a paragraph. Prior work Xiao et al. (2024) characterized this phenomenon as "attention sinks", suggesting that the initial tokens tend to attract a disproportionate share of attention. We observed it through depicting the heatmap of attention matrix and the percentile rank of the first image token in Appendix B. We conclude this as Insight 2.

**Insight 2.** *The tokens at the beginning of all image tokens receive more attention.*

The results of the motivational experiment indicate that only a few tokens are of importance, and they typically occur at the beginning of the sequence. These insights are helpful for us to address the accuracy challenge. We have verified our insights using another MLLM, InternVL-2.5. The code and detailed results are provided in our supplementary material.

## 3 SYSTEM DESIGN AND IMPLEMENTATION

### 3.1 SYSTEM OVERVIEW

This section presents the architecture of INFOBLEND as shown in Figure 5. We will start with the key components of INFOBLEND, and then introduce its workflow.

There are five key components in INFOBLEND. (1) **MLLM inference serving system** is responsible for generating output tokens in multiple steps (commonly referred to as the decode phase Zhong et al. (2024)). It also contains a scheduler that manages users' queries. The MLLM subsystem incorporates advanced techniques such as PagedAttention Kwon et al. (2023) and continuous batching Yu et al. (2022). (2) **Static Library** stores the KV cache of files uploaded by users. The files from different users are logically separated. Each user can access only his/her own files. It is relatively static, as it can only be modified by the users. This component is analogous to the static-linked code: Users refer to these files in their queries, and INFOBLEND links the KV cache of these files for the MLLM to inference. (3) **Dynamic Library** serves as the storage for multimedia references and related KV cache. This component is prepared for MRAG, for the sake of enhancing MLLM's quality and factuality. It is relatively dynamic, since the administrator of INFOBLEND can update the references periodically according to the demand of applications. This component is analogous to the dynamic-linked library: The MLLM retrieves the references during decode phase, when it determines that a retrieval is required Asai et al. (2023); Jeong et al. (2024). (4) **Retriever** is responsible for searching the relevant information based on the query. It is analogous to the relocation table when executing a program, in that the program needs the relocation table to find the address of dynamic-linked libraries. (5) **Linker** links the KV cache of multimodal information to users'

queries. Linking without accuracy degradation is an algorithm-level challenge, and we will address this challenge through selective attention mechanism in the next section.

The serving workflow of INFOBLEND is outlined as follows. ① Users upload their respective files, and INFOBLEND subsequently performs the computation of the KV cache, which is then stored in GPU memory for the purpose of serving. Concurrently, these caches are copied to disks and subsequently deleted following the expiration of their designated timeframe. ② The user submits a query, including questions and references to specific files. ③ The MLLM accesses the files according to the user's ID and references. ④ The retriever executes MRAG when triggered by the MLLM. ⑤ Linker blends the KV caches together and sends them to the MLLM as an input. ⑥ The MLLM generates an answer to respond to the user.

## 3.2 KV CACHE TRANSFER IN PARALLEL

Normally, the KV caches of files from active users are loaded on the memory of computing chip (e.g., GPU, TPU, NPU, etc.). When the stored KV caches are not on the chip, we design a parallel transfer mechanism as shown in Figure 6.

Suppose that the input includes $n$ images. INFOBLEND looks up the KV caches of these images at the beginning of processing. The KV caches of $m$ images are missing, due to deletion after expiration. The KV caches of remaining $n - m$ images are hit. Some of them are on GPU memory, while others are on CPU memory or disks. Subsequently, the computation and transfer of KV caches are executed concurrently, for the purpose of reducing the preparation time. In the case of failures when loading KV caches from disks, INFOBLEND will continue to calculate all KV caches and keep serving. The procedure falls back to inference without the existing KV cache.

Other optimization techniques such as layer-wise transfer Patel et al. (2024) and KV cache compression Liu et al. (2024c) are orthogonal to our work. They can be employed in our system as well.

## 4 IMPLEMENTATION OF SELECTIVE ATTENTION

In order to partially reuse the KV caches, we implement the selective attention mechanism. Only the selected tokens are passed into the MLLM, while they need to calculate the attention score with other tokens. We first introduce this mechanism, and then explain how to select tokens.

### 4.1 ILLUSTRATION OF SELECTIVE ATTENTION

The process is illustrated in Figure 7. $W_K$ and $W_Q$ in the figure are the parameters of MLLM. The recomputed K tensor substitutes part of the K cache for calculating the attention matrix. Here we utilize the tricky "*dummy cache*". Since the KV cache of mutable text is not saved in advance, the tokens of text must be computed. In other words, the tokens of text are part of selected tokens. The KV cache of selected tokens will be replaced, so we do not need to pre-compute the KV cache of text, and instead fill its KV cache with zeros. In this way, the selective attention mechanism is a single-step process. This is more efficient than the two-step process of *full reuse*. Our experiment in the next section exhibits the efficiency.

### 4.2 SELECTING IMPORTANT TOKENS

As analyzed in § 2.3, the attention matrix is sparse, and only 5% of tokens receive significant attention scores. In this section, we determine which tokens are important and require recomputation through a motivational experiment.

In the experiment, we compute two KV caches of a single image with different positions. Specifically, the first KV cache is created when the image is inserted before a question, while the second KV cache is created with the reverse order. We focus on the distance between the two KV caches. To find the tokens with large distance, we sort the image tokens by their K distance, and count the number of transformer layers where the token is in the top 50, as shown in Figure 8. We conclude the finding of the experiment as Insight 3.

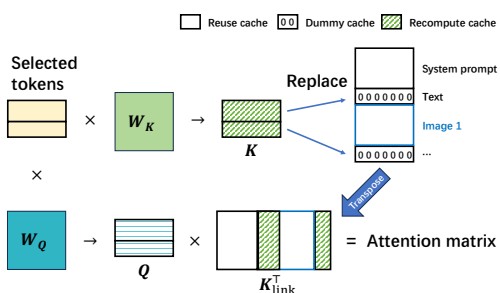

Figure 7: Illustration of selective attention. First, select the important tokens. Second, replace the respective K cache with generated $K$. Third, multiply $Q$ with $K_{\mathrm{link}}^{\top}$ to obtain the attention matrix. The V cache also undergoes the same operation. The details such as positional embedding and softmax are omitted in the figure.

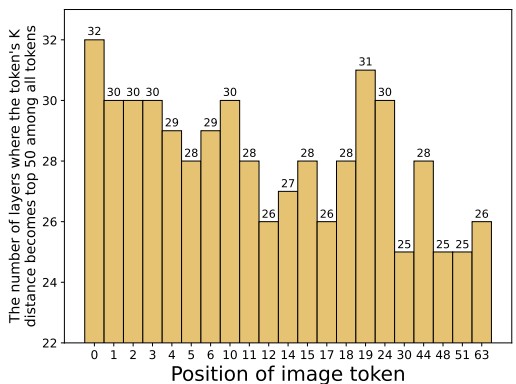

Figure 8: Important tokens. The K distance of a token is defined as the L1 distance between its two K tensors. For clarity, we only show tokens whose number is greater than 24.

**Insight 3.** *The tokens at the beginning of all image tokens exhibit a greater disparity in terms of the KV distance between reused KV tensor and recomputed KV tensor.*

In summary, the tokens at the beginning are more different (Insight 3) and receive more attention (Insight 2). These tokens absorb too much attention due to "attention sinks" and contribute negatively to the outcome. Consequently, we select all text tokens and $k$ tokens at the beginning of image tokens to be recomputed in the selective attention mechanism. We recompute them to make them realize they are not the initial tokens of the sentence anymore and stop absorbing attention, and then the accuracy can be recovered. This method is referred to as INFOBLEND-$k$. Our evaluations in the next section verify the effectiveness of INFOBLEND-$k$.

## 5 EVALUATION

In this section, we evaluate INFOBLEND in terms of response time and generation quality. We also investigate the throughput of INFOBLEND in the online scenario. The sensitivity analysis is left to Appendix C due to page limit.

### 5.1 EXPERIMENTAL SETTINGS

We select two prevalent MLLMs in the experiments: LLaVA-1.6-vicuna-7B and LLaVA-1.6-mistral-7B Liu et al. (2024b). All experiments are run on a server with 1 NVIDIA H800-80 GB GPU, 20-core Intel(R) Xeon(R) Platinum CPUs, and 100GB DRAM.

Two datasets are used in our evaluation. (1) **MMDU** Liu et al. (2024d) aims to evaluate MLLMs' abilities in multi-turn and multi-image conversations. Each conversation stitches together multiple images and sentence-level text (e.g., "Can you describe these images as detailed as possible? IMAGE#1, IMAGE#2."). (2) **SparklesEval** Huang et al. (2024) is a dataset for assessing MLLMs' conversational competence across multiple images and conversation turns. Unlike MMDU, SparklesEval integrates multiple images at word level (e.g., "Can you link the celebration occurring in IMAGE#1 and the dirt bike race in IMAGE#2 ?"). As shown in the examples, the prompts of two datasets are open questions. Previous works adopt GPT score to evaluate the quality of MLLMs' responses to the open questions Liu et al. (2024d); Huang et al. (2024). GPT score is a GPT-assisted evaluation that uses a powerful judge model (e.g., GPT-4o, Qwen, etc.) to assess the answers. We also employ this metric and their evaluation prompt, as listed in Appendix D.

We compare INFOBLEND-$k$ with three existing CC algorithms: prefix caching, full reuse, and CacheBlend Yao et al. (2025). CacheBlend is also a position-independent algorithm designed for RAG system. It recomputes $r\%$ of total tokens with largest KV deviation, so we denote it as CacheBlend-$r$. The primary focus of CacheBlend is the KV deviation, while the INFOBLEND's

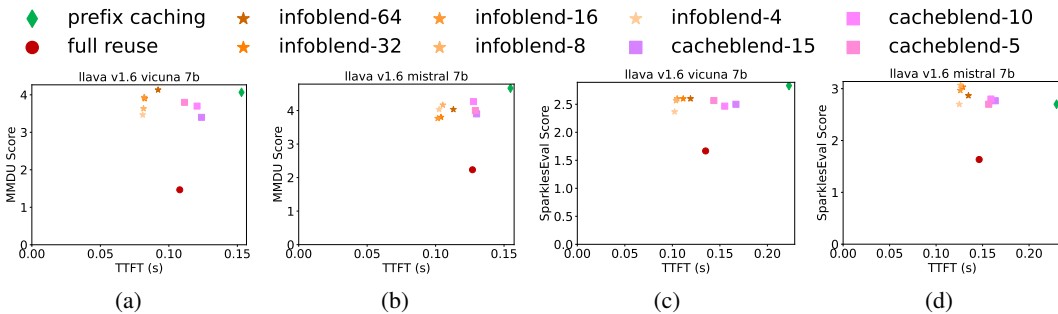

Figure 9: Comparison of TTFT (↓ Better) and Score (↑ Better) using different models on different datasets. Figures (a)(c) show the results of LLaVA-1.6-vicuna-7B, while Figures (b)(d) show the results of LLaVA-1.6-mistral-7B. The label of y-axis indicates the dataset used.

selection process involves the identification of tokens that exhibit both high attention scores and significant KV deviation. We implement the four CC algorithms based on vLLM 0.9.0 Kwon et al. (2023).

## 5.2 EFFECTIVENESS OF INFOBLEND

Based on vLLM offline inference, we compare the performance of all algorithms. Specifically, we process all requests sequentially and evaluate their generation quality and processing time for prefill. The workflow initiates with the precomputation of the relevant KV cache for images. Subsequently, we send the user's query along with the cache_ids of the images to the serving system. Prefix caching will process the query with the KV cache of system prompt only. INFOBLEND concatenates the dummy cache and stored cache, and computes the first output token using selective attention mechanism in single step. Full reuse and CacheBlend first compute the KV cache of text, and then produce the first output token with the concatenated KV cache. We record the processing time of the algorithms and finally score for each response.

Figure 9 presents the experimental results of all algorithms across different models and datasets. The results indicate that INFOBLEND consistently outperforms CacheBlend in terms of both TTFT and score across various configurations. INFOBLEND-32 reduces TTFT by up to 54.1% while maintaining a loss of score within 13.6% compared to prefix caching. Additionally, it is clear that IN-FOBLEND exhibits a slight decrease in TTFT compared to full reuse, since INFOBLEND is a single-step process. Compared to other algorithms, INFOBLEND achieves the best trade-off between TTFT and score.

## 5.3 THROUGHPUT OF INFOBLEND IN THE ONLINE SCENARIO

To assess INFOBLEND's latency and throughput performance, we leverage vLLM's OpenAI-compatible API server to simulate real-world user request patterns. We first sample dozens of instances from MMDU and pre-generate KV caches for their images. Subsequently, we simulate user request behavior by repeatedly sending the user queries along with the cache_ids of these samples at a specified request rate over a period of time. Through varying the request rate, we measure the latency and throughput across different experimental conditions.

Figure 10 presents a comparison of latency and throughput in the online scenario. As the request rate increases, the TTFT of serving systems rises significantly. This phenomenon can be attributed to the fact that, in high-demand scenarios, requests are required to queue, thereby delaying the processing of individual requests. As the request rate increases, the throughput of serving systems first rises linearly, then falls, and finally stabilizes around a fixed value. The throughput rises linearly because the low request rate cannot saturate the system. When the request rate exceeds the capability of the system, the throughput falls and converges to a fixed value. INFOBLEND consistently outperforms CacheBlend and prefix caching in terms of TTFT and throughput. Compared to CacheBlend, In-foBlend achieves up to 73.5% reduction in TTFT and 2.0× improvement in throughput. This gap increases as the request rate rises.

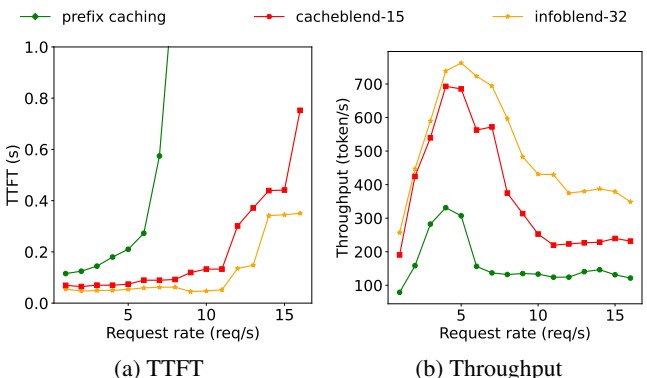

Figure 10: The performance of INFOBLEND as the request rate increases. The throughput is defined as the rate of generating output tokens.

## 6 RELATED WORK

**LLM Serving Optimizations.** Several serving systems emerged in the past year. vLLM Kwon et al. (2023) is a pioneering work in this space featuring PagedAttention for higher throughput. SGLang Zheng et al. (2024) is another serving system featuring a novel frontend language and a backend runtime. Aside from full-systems, there are also many scheduling optimizations such as disaggregated prefill and decode Zhong et al. (2024); Hu et al. (2024a;b); Patel et al. (2024), continuous batching Yu et al. (2022), multi-lora Sheng et al. (2023); Li et al. (2024), etc.

**Context Caching (CC).** Context Caching (CC) has two main categories. The first is Position-dependent caching, which can be further divided into prefix-based caching Yu & Li (2023); Zheng et al. (2024) and modular caching Gim et al. (2024). By mid-2024, vendors such as Kimi Moonshot (2024) and Gemini Google (2024a) began incorporating explicit CC features into their systems. The second category is Position-Independent Caching (PIC). To the best of our knowledge, CacheBlend Yao et al. (2025) represents the first work addressing aspects of PIC, while EPIC Hu et al. (2025) formally defines the PIC problem and advances the state-of-the-art one step forward. However, these two pieces of work focus mainly on text-based applications, and are inefficient in the context of interleaved text and images. In order to prevent triggering MLLM engine twice, we design selective attention mechanism to process the prefill phase of queries in single step. Our work is the first to explore a solution to the PIC problem in the multi-modality field.

**Retrieval-Augmented-Generation (RAG).** RAG is a technique that combines retrieval-based methods with generation-based models to improve the performance of LLMs. It aims to address the limitations of purely generative models, which can sometimes lack factual accuracy or struggle with long-context understanding Li et al. (2022); Jin et al. (2024); Gao et al. (2023); Jeong et al. (2024); Ram et al. (2023); Mao et al. (2020). For example, Adaptive-RAG Jeong et al. (2024) develops a dynamic framework to select the most suitable strategy for LLMs to deal with queries based on their complexity. To classify queries of different complexity, Adaptive-RAG trains a small model as a classifier to predict the complexity of queries. We also incorporate the RAG into INFOBLEND, and enhance its efficiency through the reuse of KV cache.

## 7 CONCLUSION

In this paper, we present INFOBLEND to store and reuse KV caches of multimodal information without positional restriction. We design parallelized KV cache transfer and selective attention mechanism to address the system-level and algorithm-level challenges respectively. We have analyzed the characteristic of image tokens in detail, and select the important tokens with the insights. Experimental results across different models and datasets illustrate that INFOBLEND reduces TTFT by 54.1% with negligible or no quality degradation, and doubles the throughput in the scenario of online services. Our solution is applicable to large number of images as well.

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

# A  Transformer Primer

The attention-based transformer architecture Vaswani et al. (2017) underpins most large language models (LLMs) today. A typical LLM comprises multiple transformer layers, each containing an attention module. This module maps input vectors to query, key, and value vectors, which it then combines to produce an output vector. Specifically, when an attention module receives input vectors, or a sequence of hidden states $(x_1, x_2, ...x_n) \in \mathbb{R}^{n \times d}$, where $n$ is the number of tokens in the sequence, and $d$ is the hidden dimension, it computes the key, query, and value vectors for each token as follows:

$$k_i = W_k x_i, q_i = W_q x_i, v_i = W_v x_i.$$

where $W_k, W_q$, and $W_v$ are learnable weight matrices. The attention module then computes the attention score between tokens as follows:

$$a_{ij} = \frac{\exp(q_i^\top k_j / \sqrt{d})}{\sum_{t=1}^{i} \exp(q_i^\top k_t / \sqrt{d})}$$

Finally, the attention module computes the output vectors as weighted sums of the value vectors:

$$o_i = \sum_{j=1}^{i} a_{ij} v_j$$

The output of the attention module can either serve as the input to the next layer or act as the final output.

## A.1  Multimodal Transformer

While the original transformer architecture Vaswani et al. (2017) was initially developed for natural language processing (NLP) tasks, its attention-based mechanism has been successfully extended to a wide range of multi-modal applications. In the multi-modal setting, the key challenge is to effectively combine information from heterogeneous modalities, each with distinct structures and representations. Existing approaches address this by using modality-specific encoders. For example, in vision-language tasks, one encoder processes image features (often using convolutional neural networks or vision transformers), while another processes textual input, such as captions or queries. These encoders project modality-specific features into a shared embedding space, enabling the transformer model to process embeddings from different modalities uniformly.

## A.2  Autoregressive Generation & KV Cache

LLMs generate tokens autoregressively, predicting one token at a time based on the input. This process involves two phases: the prefill phase and the decode phase. In the **prefill** phase, LLMs process the entire input prompt $(x_1, x_2, \ldots, x_n)$, computing and caching the key and value vectors for each token. This phase can be slow for long inputs, and the time to generate the first token is measured by the Time-to-First-Token (TTFT) metric. In the **decode** phase, LLMs generate one token at a time. The model computes the probability of the next token $x_{n+1}$, selects the most likely token, and appends its key and value vectors to the KV cache.

The KV cache Pope et al. (2023), which stores the key and value vectors computed by the attention module, accelerates generation by enabling intra-request and inter-request reuse. First, within a single request, the cache speeds up token generation by allowing the model to reuse cached KV vectors, thus only processing the new token instead of recalculating vectors for the entire sequence. Second, when a new request shares a prefix with a previous one, the KV cache for the prefix can be reused, thereby accelerating the prefill phase of the new request.

## A.3  Context Caching

To better use KV cache, existing approaches propose context caching (CC) that exploits inter-request dependency to reuse KV cache across inference requests to avoid repeated computation of the same prompt tokens Kwon et al. (2023); Zheng et al. (2024); Yao et al. (2025); Hu et al. (2025). There are two types of context caching. First, in **prefix-based context caching**, only the initial portion

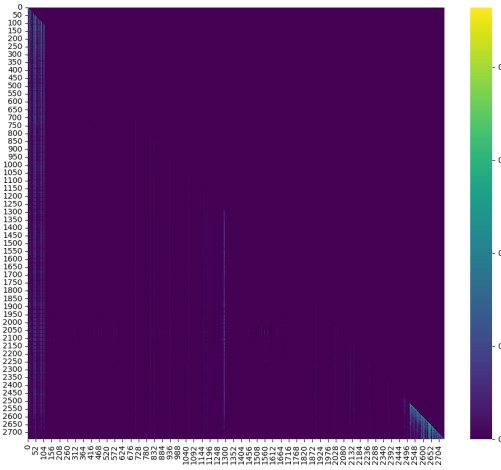

Figure 11: The attention heatmap of an example with two images. Tokens from 2512 to 2731 are output tokens.

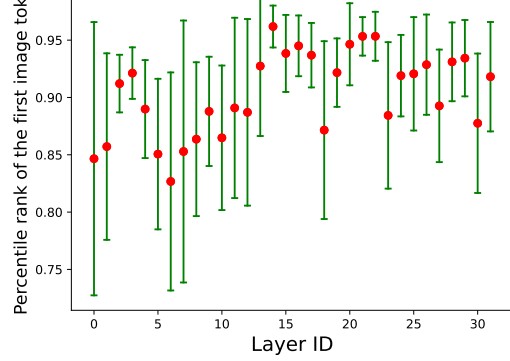

Figure 12: The percentile rank of the first image token in all tokens across different layers. The red dots are the mean value, while the green error bars are the standard deviation across different heads.

of the sequence is cached and reused. Almost all existing CC offerings and designs are prefix-based Kwon et al. (2023); Zheng et al. (2024); Hu et al. (2024a); Zhong et al. (2024). However, in this approach, if the prefix differs slightly (e.g., one or two words are different at the beginning), the entire cache cannot be reused, forcing the model to recompute all the key-value pairs for that request. This inefficiency can be particularly problematic in cases where many requests share a substantial amount of overlapping context but differ slightly in their initial tokens (e.g., in RAG). Second, the **position-independent context caching** addresses the limitation of prefix-based caching by enabling KV cache reuse across requests, even when token sequences differ, without being constrained by the shared prefix. The position-independent approach was a new concept and was explored in two CacheBlend Yao et al. (2025) and EPIC Hu et al. (2025) on Natural Language Processing (NLP) tasks. Our work is the first to explore a solution to the PIC problem in the multi-modality field.

## B  INVESTIGATION OF THE IMPORTANCE OF INITIAL IMAGE TOKENS

This appendix provides detailed evidence to illustrate the importance of initial image tokens.

### B.1  ATTENTION HEATMAP

We use the example in Figure 1 with LLaVA-1.6-vicuna-7B to generate the attention scores by three steps. First, we discard the negative attention scores, i.e., setting them to 0. Second, the attention scores are normalized through min-max normalization. Third, we calculate the average value of 32 heads in the first transformer layer, and show the value in Figure 11. It is observed that the beginning tokens of two image, token 109 and token 1294, receive more attention scores from other tokens.

### B.2  PERCENTILE RANK

Figure 12 shows the distribution of the percentile rank of the first image token in all tokens across different layers. In each layer, we collect the percentile rank of the attention score between the first image token and the first output token across different heads. It is evident that the first image token receives more attention score than 80% of all tokens, in most layers and heads.

## C  ADDITIONAL RESULTS: SENSITIVITY ANALYSIS

In order to achieve a more profound comprehension of INFOBLEND, a subsequent analysis is necessary to ascertain how the number of images impacts overall performance. We divide the dataset

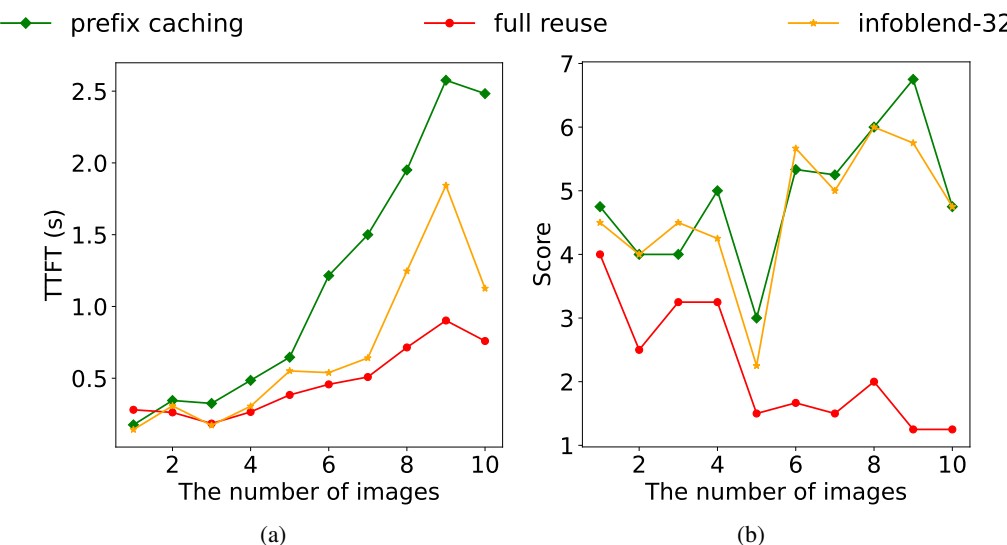

Figure 13: The performance of INFOBLEND as the number of images increases. For clarity, we only present the results of INFOBLEND-32. Other variants of INFOBLEND show similar patterns.

of MMDU into 10 groups in terms of the number of images. We evaluate the TTFT and score of INFOBLEND and baselines on each group. The average value of results are shown in Figure 13. The TTFT of INFOBLEND is consistently shorter than that of prefix caching. When the number of images is 10, INFOBLEND achieves 54.7% reduction in TTFT. Furthermore, the performance of INFOBLEND remains unaffected by the number of images, exhibiting negligible or no accuracy degradation.

## D    EVALUATION PROMPT

We evaluate the answers to open questions with the assist of chatGPT. The evaluation prompt imported from MMDU Liu et al. (2024d) is as follows.

"You are an assistant skilled at evaluating the quality of creative text. Please act as an impartial judge and evaluate the quality of the response provided by an AI assistant to the user question displayed below. You'll need to assess the response on the following dimensions: Creativity, Richness, Visual Perception, Logical Coherence, Answer Accuracy and Image Relationship Understanding. We will provide you with a creative question and the AI model's response and a reference answer for your evaluation. As you begin your assessment, follow this process: 1. Evaluate the AI model's answers on different dimensions, pointing out its strengths or weaknesses in each dimension and assigning a score of 1 to 10 for each. 2. Finally, based on the assessments across dimensions, provide an overall score of 1 to 10 for the AI model's response. 3. Your scoring should be as stringent as possible and follow the scoring rules below:

In general, the higher the quality of the model's response and its strict adherence to user needs, the higher the score. Responses that do not meet user needs will receive lower scores.

Scoring rules: Creativity: Scores 1-2 when there is no innovation or uniqueness in the content. Scores 3-4 when providing partially original content but with low creative quality. Scores 5-6 when mostly creative but lacks significant novelty, with moderate quality. Scores 7-8 when having novelty and high-quality content. Scores 9-10 when highly novel and of exceptional quality compared to the reference answer.

Richness: Scores 1-2 when lacking depth and breadth, with very limited information. Scores 3-4 when limited in depth and breadth, with fewer explanations and examples, showing low diversity. Scores 5-6 when limited in depth and breadth but provides basic necessary information. Scores 7-8

when providing depth and useful additional information. Scores 9-10 when providing exceptional depth, breadth, and high diversity compared to the reference answer.

Visual Perception: Scores 1-2 when the description of the visual information in the image contains errors or is significantly inconsistent with the content of the image. Scores 3-4 When the description of the visual information in the image reflects only a small amount of the image's information and contains some errors. Scores 5-6 when the description of the visual information in the image includes the basic information of the image but contains minimal information. Scores 7-8 when the description of the visual information in the image matches the image well and is rich in content, providing a substantial amount of information about the image. Scores 9-10 when the description of the visual information in the image not only matches the image but also is more detailed and informative compared to the reference answer, providing more information about the image.

Logical Coherence: Scores 1-2 when entirely incoherent, lacking any logic, and not matching the question or known information. Scores 3-4 when somewhat coherent but with many logical errors or inconsistencies. Scores 5-6 when mostly coherent, with few errors, but may struggle to maintain complete coherence in complex situations. Scores 7-8 when excellent logical handling, very few errors. Scores 9-10 when flawless logic, impeccable in handling complexity, and significantly higher logical coherence compared to the reference answer.

Answer Accuracy Scores 1-2 when the answer is significantly inconsistent with the question or contains obvious errors. Scores 3-4 when the answer is partially correct but contains some errors or is incomplete. Scores 5-6 when the answer is basically correct but lacks details or is not sufficiently detailed. Scores 7-8 when the answer is accurate and detailed, fully corresponding to the question. Scores 9-10 when the answer is not only accurate and detailed but also provides additional useful information, exceeding expectations.

Image Relationship Understanding: Scores 1-2 when there are significant errors or confusion in distinguishing and describing different images, unable to correctly identify and relate the content of the images. Scores 3-4 when the description of different images reflects only minimal distinguishing information, contains some errors and confusion, and fails to clearly differentiate and relate the images. Scores 5-6 when the description of different images includes basic distinguishing information, is able to correctly identify and relate the images in a basic manner, but the information provided is minimal and lacks detail. Scores 7-8 when the description of different images is accurate and detailed, clearly distinguishing and relating the images, with rich content that points out the main commonalities and differences between the images. Scores 9-10 when the description of different images is not only accurate and detailed but also provides richer information and analysis, clearly distinguishing and relating the images, more comprehensively pointing out the commonalities and differences between the images compared to the reference answer.

Overall Score: Scores 1-2 when irrelevant to the question, factually incorrect, or generates harmful content. Scores 3-4 when no serious errors, mostly harmless, but of low quality and does not meet requirements. Scores 5-6 when basically meeting requirements but performing poorly in some dimensions, with moderate quality. Scores 7-8 when performing well in all dimensions. Scores 9-10 when fully addressing user questions and all requirements, significantly surpassing the reference answer.

Please remember, you must evaluate and explain before scoring. After your explanation for each dimension, add the score for that dimension. Finally, at the end of your response, in the format of the dictionary (including brackets), return all your scoring results, ensuring your scores are integers:

{'Dimension One': Score, 'Dimension Two': Score, ..., 'Overall Score': Score}, for example: {'Creativity': 9, 'Richness': 6, ..., 'Overall Score': 7}."

