# OpenReview forum: "InfoBlend: Storing and Reusing KV Caches of Multimodal Information without Positional Restriction"
_ICLR.cc/2026/Conference — Submitted to ICLR 2026_

### Official Review · Reviewer_aAof · 2025-10-27

**Soundness:** 3
**Presentation:** 3
**Contribution:** 3
**Rating:** 6
**Confidence:** 2

**Summary:**

1.  **System Design:** Stores KV caches on disk, loads/computes them in parallel during inference, and uses static/dynamic libraries for different cache types (user uploads vs. MRAG content).
2.  **Selective Attention Mechanism:** To mitigate accuracy loss from reusing positionally shifted KV caches, INFOBLEND recomputes all text tokens and the first 'k' tokens of each multimodal input's KV cache (INFOBLEND-k). This is based on the insight that initial tokens act as "attention sinks" and their KV values are most affected by positional changes. This is implemented efficiently as a single-step prefill process using a "dummy cache" technique.

**Strengths:**

1.  **Problem Significance:** Addresses a clear and growing bottleneck (prefill latency) in MLLM serving, particularly for complex multimodal inputs like interleaved documents or MRAG results, which hinders user experience and scalability.
2.  **Novel Approach:** Proposes position-independent caching (INFOBLEND) for MLLMs, moving beyond the limitations of standard prefix caching. This is a necessary step for efficiently handling dynamic multimodal contexts.
3.  **Insightful Heuristic:** The selective attention mechanism (recomputing the first k image tokens) is well-motivated by empirical analysis of attention patterns (sparsity, initial token sinks) in MLLMs. This provides a principled way to trade off computation and accuracy.
4.  **Practical System Design:** Considers system-level challenges like large cache sizes (disk storage) and latency hiding (parallel loading/computation). The single-step prefill implementation using dummy caches enhances efficiency over two-step approaches.
5.  **Strong Empirical Results:** Demonstrates significant improvements in TTFT (up to 54.1% reduction) and throughput (up to 2.0x increase) compared to prefix caching and CacheBlend, while maintaining relatively high generation quality.

**Weaknesses:**

1.  **Accuracy Trade-off:** While INFOBLEND significantly improves quality over "full reuse", there is still an accuracy degradation compared to "prefix caching" (up to 13.6% score loss mentioned, though Figure 9 suggests the best INFOBLEND variants are closer). The paper could discuss the practical implications of this trade-off more – is this level of score drop acceptable for production systems?
2.  **Heuristic Generalizability:** The strategy of recomputing the first 'k' tokens is based on observations in LLaVA models. While plausible, its universal applicability across different MLLM architectures (e.g., those with different vision encoders or cross-attention mechanisms) or modalities (video, audio) is not fully validated.
3.  **GPT Score Reliability:** The evaluation relies on GPT scores for generation quality. While common, this metric can be noisy and potentially biased. Including human evaluations or other automated metrics (e.g., ROUGE, BLEU for relevant tasks, or specific VQA accuracy if applicable) could strengthen the quality assessment.
4.  **Comparison Details:** The comparison with CacheBlend mentions CacheBlend focuses on KV deviation while INFOBLEND incorporates attention insights. A more direct comparison of *why* INFOBLEND's heuristic is better (e.g., does KV deviation correlate poorly with attention sinks?) would be insightful.

**Questions:**

1.  Regarding the accuracy trade-off: Section 5.2 mentions up to 13.6% score loss for INFOBLEND-32 vs. prefix caching, but Figure 9 seems to show INFOBLEND variants achieving scores very close to prefix caching. Could you clarify the extent of the quality degradation for the optimal INFOBLEND-k variants and discuss the scenarios where this trade-off might be more or less acceptable?
2.  How was the value of 'k' (the number of initial image tokens to recompute) determined for INFOBLEND-k? Is there a principled way to select 'k', perhaps dynamically based on the model, modality, or context length, rather than using fixed values (4, 8, 16, 32, 64)? How sensitive are TTFT and quality to the choice of 'k'?
3.  The analysis focuses on image tokens. Does the "attention sink" phenomenon and the effectiveness of recomputing initial tokens apply similarly to other modalities like video frames or audio chunks when used in an interleaved manner?
4.  Could you elaborate on why recomputing the first k tokens based on attention sink insights leads to better performance (both speed and quality) compared to recomputing tokens based on KV deviation, as done in CacheBlend?

---

> ### Author Response · Authors · 2025-11-20
>
> We appreciate your valuable feedback. We have read the review carefully and supplemented necessary clarifications following your recommendation. Here is our response to your questions:
>
> ## Question 1 and Weakness 1
>
> > Could you discuss the scenarios where this trade-off might be more or less acceptable? The paper could discuss the practical implications of this trade-off more – is this level of score drop acceptable for production systems?
>
> INFOBLEND finds application in latency-sensitive scenarios such as game AI, voice assistant, and simultaneous interpretation. In these scenarios, users demand a rapid response and tolerate imperfect answers. In terms of generation quality, INFOBLEND generally outperforms CacheBlend.
>
> > Could you clarify the extent of the quality degradation for the optimal INFOBLEND-k variants?
>
> If $k\rightarrow\infty$, INFOBLEND fully recomputes all KV caches, so there is no quality degradation. The more tokens we recompute, the higher the accuracy. The evaluations show that Infoblend achieves scores close to prefix caching when $k\geq 32$.
>
> ## Question 2
>
> > How was the value of 'k' (the number of initial image tokens to recompute) determined for INFOBLEND-k? Is there a principled way to select 'k', perhaps dynamically based on the model, modality, or context length, rather than using fixed values (4, 8, 16, 32, 64)?
>
> The choice of parameter $k$ is determined by the block size of PagedAttention [1]. PagedAttention stores the KV caches of tokens block by block. One block contains $n$ tokens. The number $n$ is called the block size. PagedAttention employs the paging technique to manage the physical and logical blocks, thereby reducing the memory fragmentation. The value of parameter $k$ is the multiple of the block size, and 4, 8, 16, 32, 64 are common values of block size. We decide to be guided by the block size since we can update the KV caches of all tokens in a block, achieving clarity of implementation.
>
> Thank you for presenting this valuable question. We will incorporate the clarifications in the next draft.
>
> [1] Kwon, W., Li, Z., Zhuang, S., Sheng, Y., Zheng, L., Yu, C. H., Gonzalez, J. E., Zhang, H., and Stoica, I. Efficient memory management for large language model serving with pagedattention. In Proceedings of the ACM SIGOPS 29th Symposium on Operating Systems Principles, 2023.
>
> ## Question 3 and Weakness 2
>
> > Does the "attention sink" phenomenon and the effectiveness of recomputing initial tokens apply similarly to other modalities like video frames or audio chunks when used in an interleaved manner?
>
> The MLLM is a combination of a multimodal encoder and an LLM backbone. The “attention sink” phenomenon derives from the LLM backbone, and INFOBLEND is designed to avoid the initial tokens absorbing too much attention. So INFOBLEND works whatever the encoder is. We also have several small-scale experiments on videos showing the effectiveness of INFOBLEND.
>
> ## Question 4 and Weakness 4
>
> > Could you elaborate on why recomputing the first k tokens based on attention sink insights leads to better performance (both speed and quality) compared to recomputing tokens based on KV deviation, as done in CacheBlend?
>
> 1. We first notice that among the tokens recomputed by CacheBlend, the initial tokens are almost always included.
>
> 2. Further analysis of the initial tokens of each chunk reveal the attention sink problem: they absorb too much attention away, preventing later decoding tokens to attend to relevant information. Recomputing them releases this part of "sink attention" and then the accuracy can be recovered.
>
> 3. The selection strategy of CacheBlend ignores the attention score, so it needs to select $rN(\approx 350)$ tokens to cover tokens with high attention scores and recover accuracy ($r$ is the recomputation ratio and $N$ is the number of image tokens). This is inefficient compared to INFOBLEND (64 tokens per image). We supplement the result of CacheBlend-1 in the following table, which shows that the accuracy of CacheBlend is unacceptable when $r$ is small.
>
> Table 1. Percentiles of MMDU score (LLaVA-1.6-vicuna-7B)
> |Algorithms|10%|50%|90%|
> |---|---|---|---|
> |Prefix caching|2.9|4.0|7.0|
> |CacheBlend-1|1.0|2.7|4.6|
> |CacheBlend-15|2.0|3.0|5.1|
> |InfoBlend-64|2.9|4.0|6.5|
>
> To conclude, CacheBlend merely focuses on numerical equivalence and tries to find the KV caches that lead to the largest deviation, without understanding what is behind these numerical deviations. We, however, pin down the root cause of this deviation and leverage this new cognitive to reduce redundant computation.
>
> Thank you again for your insightful review. We will incorporate the clarifications and address the issues in the next draft. We would greatly appreciate it if you could raise your score if you find the answers helpful.
>
> Best regards.

---

### Official Review · Reviewer_a8xw · 2025-10-27

**Soundness:** 3
**Presentation:** 2
**Contribution:** 3
**Rating:** 4
**Confidence:** 3

**Summary:**

INFOBLEND enables position independent kv cache reuse for multimodal LLMs. While previous work like, CacheBlend recomputes tokens with the largest KV deviation, INFOBLEND also considers attention importance to selectively recompute only critical tokens, mainly at the start of image sequences. INFOBLEND uses a single-step selective attention process instead of CacheBlend’s two-step recomputation, supports parallel KV loading for multimodal data, and achieves higher throughput and lower latency.

**Strengths:**

1. Well motivated problem about multimodal KV cache reuse.
2. Good results on 4 model-dataset pairs, comparison with previous work.
3. Support for multimodal by extending into disk storage and related system optimizations around them.
4. Novel combination of attention sink and kv deviation.

**Weaknesses:**

1. Comparison between this and CacheBlend not clear enough.
2. Not sure what is new about multimodal. The delta mentioned with previous work is that there are attention sink considerations in infoblend. However, why is this multimodal specific?
3. Different rates of Infoblend seem to be not on a tradeoff frontier curve but rather a cluster of points. There should be difference in time vs accuracy. Why isn't there difference?
4. Can we report on bigger models or more sota methods like qwen-vl? Lack of discussion for these newer models

**Questions:**

1. How to set hyperparameters in infoblend? aka how to tune the delay-accuracy tradeoff curve?
2. Questions in the weakness section.

---

> ### Author Response · Authors · 2025-11-20
>
> We appreciate your valuable feedback. We have read the review carefully. We will clarify our unique contributions and incorporate this clarification in the next draft.
>
> ## Weakness 1 & 2
>
> > Comparison between this and CacheBlend not clear enough.
> > Not sure what is new about multimodal. The delta mentioned with previous work is that there are attention sink considerations in infoblend. However, why is this multimodal specific?
>
> Thank you for pointing out the similarity between CacheBlend and INFOBLEND. We will highlight the unique challenge of multimodality here.
>
> 1. CacheBlend is not efficient enough from a system perspective. **The multimodal context is interleaved with mutable text, which is different from the scenario of RAG. CacheBlend assumes that all chunks are concatenated consecutively, without considering interleaved variable text.** The KV caches of images are stored, while those of text are not. Therefore, CacheBlend first computes the KV caches of text, and then reuses the KV caches of text and images. In order to prevent triggering MLLM engine twice, we design the selective attention as shown in Figure 7. The evaluation results illustrate its efficiency. We emphasize this issue of existing papers in Lines 92-95 and Lines 192-195.
>
> 2. In terms of selection strategy, CacheBlend merely focuses on numerical equivalence and tries to find the KV caches that lead to the largest deviation, without understanding what is behind these numerical deviations. We, however, pin down the root cause of this deviation (attention sink) and leverage this new cognitive to reduce redundant computation.
>
> 3. Last but not least, the computation overhead of INFOBLEND is theoretically less than that of CacheBlend. Suppose that the number of image tokens is $N$. Initially, computing the attention score for $N$ tokens has an $O(N^2)$ complexity.
> - INFOBLEND computes $k$ tokens for each image (Figure 7), leading to $O(kN)$ complexity.
> - CacheBlend computes $rN$ tokens for each image, leading to $O(rN^2)$ complexity.
>
> ## Weakness 3 and Question 1
>
> > Different rates of Infoblend seem to be not on a tradeoff frontier curve but rather a cluster of points. There should be difference in time vs accuracy. Why isn't there difference?
>
> - **Time**: As shown in Figure 9, the TTFT of Infoblend grows as $k$ increases. This is because the amount of computation increases. The recomputation cost of INFOBLEND is so low that even if we increase the number of tokens to recompute, the TTFT remains similar.
> - **Accuracy**: Generally speaking, the accuracy of Infoblend grows as $k$ increases. Several points of Figures 9(b)(d) deviate from this law due to the difficulty of scoring open-ended questions. We find that the answers of different Infoblend variants are similar, so their scores on open questions are similar. In addition, as we increase the number of tokens to recompute, the accuracy stables and remains comparable to CacheBlend because we have pinned down the root cause of accuracy degradation in PIC, thus recovering the same level of accuracy with minimal computations. And if we try to compute more tokens than the first few tokens of each chunk (the root cause), there will be marginal gains.
>
> > How to set hyperparameters in infoblend? aka how to tune the delay-accuracy tradeoff curve?
>
> As $k$ increases, the latency and accuracy grows generally. The evaluations show that Infoblend achieves scores close to prefix caching when $k\geq 32$, so we recommend using Infoblend-32 or Infoblend-64.
>
> ## Weakness 4
>
> > Can we report on bigger models or more sota methods like qwen-vl?
>
> Infoblend recomputes the initial tokens to prevent them from absorbing too much attention (“attention sink”). The “attention sink” phenomenon emerges on bigger and newer models such as InterVL (analyzed in supplementary materials), so Infoblend can be generalized to these models. Though we do not have resources to finish full evaluations on big models, we have tried on big models and inspect the same atttention sink phenomenon.
>
> Thank you again for your insightful review. We are certain that your review will help us improve our paper. We sincerely invite you to re-evaluate the paper. Please let us know if the response addresses the weakness.
>
> Best regards.

---

### Official Review · Reviewer_ozB4 · 2025-10-29

**Soundness:** 2
**Presentation:** 2
**Contribution:** 2
**Rating:** 4
**Confidence:** 5

**Summary:**

Paper presents INFOBLEND, a position-independent KV Cache reuse system across multi-modal inputs. Evaluation across multiple MLLMs and datasets illustrate that INFOBLEND saves TTFT by 54.1% compared to prefix caching, with an accuracy loss within 13.6%

**Strengths:**

* Identifies a problem that multi-modal inputs make it harder to naively reuse KV caches due to the interleaving of text and images, making the space of combined inputs much larger.
* Shows important illustrations of the KV trends for KV Caches of images, beyond the traditional KV Caches of text tokens.

**Weaknesses:**

* I am not sure the paper adds too much technical novelty beyond applying position-independent KV Cache reuse beyond applying it to a slightly newer domain. The authors claim in their evaluation, *"CacheBlend is also a position-independent algorithm designed for RAG system. The primary focus of CacheBlend is the KV deviation, while the INFOBLEND’s selection process involves the identification of tokens that exhibit both high attention scores and significant KV deviation"*  It is not clear what exactly INFOBLEND is adding here, as *"identification of tokens that exhibit high attention scores"* leads to higher KV deviation from the average values. It seems to be the same metric viewed from two different angles.

* The authors claim *"INFOBLEND-32 reduces TTFT by up to 54.1% while maintaining a loss of score within 13.6% compared to prefix caching",* This seems like a significant quality drop, which makes me concerned about the generalisation of this method to newer data, which could have worse degradation.

**Questions:**

* How exactly is the partial reuse applied on the image tokens? I'd encourage the authors to clarify this, as it seems to be a major motivation behind the design, but I couldn't find a clear explanation.
* There are many optimizations proposed in the paper, such as multi-tier KV storage and parallel loading. It would be useful to see a breakdown of where the gains in INFOBLEND are coming from. Is it mainly due to these optimizations, or does it come from the technical novelty of KV reuse for multi-modal data?
* Figure 10 could have a more meaningful comparison if the value of k were similar for Cacheblend and INFOBLEND. I'd encourage the others to add this as currently the gains are not clearly visible and are hard to compare.

---

> ### Author Response · Authors · 2025-11-20
>
> We appreciate your valuable feedback. We have read the review carefully and supplemented the necessary clarifications following your recommendation.
>
> ## Weakness 1
>
> > It is not clear what exactly INFOBLEND is adding here, as "identification of tokens that exhibit high attention scores" leads to higher KV deviation from the average values. It seems to be the same metric viewed from two different angles.
>
> Let's start with the definition of KV deviation and attention score. Let the cached and recomputed key tensor of token $i$ be $k_i^\text{old}, k_i^\text{new}$ respectively. Then the KV deviation is defined as $\|k_i^\text{old}-k_i^\text{new}\|$, and the attention score with respect to query tensor $q$ is defined as $\text{softmax}(\frac{qk_1^\text{new}}{\sqrt{d}}, \frac{qk_2^\text{new}}{\sqrt{d}},\cdots,\frac{qk_N^\text{new}}{\sqrt{d}})$, where $d$ is the hidden dimension.
>
> According to the definitions, a high KV deviation does not necessarily translate to a high attention score. The selection strategy of CacheBlend ignores the attention score, so it needs to select $rN(\approx 350)$ tokens to recover accuracy ($r$ is the recomputation ratio and $N$ is the number of image tokens). This is inefficient compared to INFOBLEND (64 tokens per image). We supplement the result of CacheBlend-1 in the following table, which shows that the accuracy of CacheBlend is unacceptable when $r$ is small.
>
> Table 1. Percentiles of MMDU score (LLaVA-1.6-vicuna-7B)
> |Algorithms|10%|50%|90%|
> |---|---|---|---|
> |Prefix caching|2.9|4.0|7.0|
> |CacheBlend-1|1.0|2.7|4.6|
> |CacheBlend-15|2.0|3.0|5.1|
> |InfoBlend-64|2.9|4.0|6.5|
>
> To conclude, CacheBlend merely focuses on numerical equivalence and tries to find the KV caches that lead to the largest deviation, without understanding what is behind these numerical deviations. We, however, pin down the root cause of this deviation and leverage this new cognitive to reduce redundant computation.
>
> ## Weakness 2
>
> > This seems like a significant quality drop, which makes me concerned about the generalisation of this method to newer data, which could have worse degradation.
>
> We acknowledge the 13.6% quality drop shown in Figure 9b on the MMDU dataset. However, Figure 9d shows an improvement in quality on the SparklesEval dataset. This is due to the difficulty of scoring open-ended questions. We find that the answers of INFOBLEND and prefix caching are similar, so their scores on open questions are similar.
>
> INFOBLEND finds application in latency-sensitive scenarios such as game AI, voice assistant, and simultaneous interpretation. Furthermore, INFOBLEND generally outperforms CacheBlend. Since PIC is a ground-breaking concept, we dedicate ourselves to increase the accuracy to the level as the original attention in future work.
>
> ## Question 1
>
> > How exactly is the partial reuse applied on the image tokens?
>
> We will provide the following pseudocode of selective attention to clarify the algorithm. Figure 7 in the paper also shows the partial reuse pipeline.
>
> ```python
> """
> Input:
>     q (tensor): the recomputed q
>     k (tensor): the recomputed k
>     v (tensor): the recomputed v
>     kv_cache (`tuple` of tensor): the stored KV cache (already concatenated with dummy cache)
>     attn_metadata (`dict`): other parameters, including selected token ids.
> Output:
>     o (tensor): the output of attention algorithm
> """
> def attn_fwd_with_pic(q, k, v, kv_cache, attn_metadata):
>     key_cache, value_cache = kv_cache
>     # get the indices of important tokens, which are computed after tokenization
>     imp_indices = attn_metadata["imp_indicies"]
>     # replace the old KV cache with recomputed one
>     key_cache[imp_indices] = k
>     value_cache[imp_indices] = v
>     k = key_cache
>     v = value_cache
>
>     # attention algorithm
>     o = attn_fwd_func(q, k, v, attn_metadata)
>     return o
> ```
>
> ## Question 2
>
> > Is it mainly due to these optimizations, or does it come from the technical novelty of KV reuse for multi-modal data?
>
> The latency reduction comes mainly from the selective attention algorithm. Suppose that the number of image tokens is $N$. Initially, computing the attention score for $N$ tokens has an $O(N^2)$ complexity. INFOBLEND computes $k$ tokens for each image (Figure 7), leading to $O(kN)$ complexity. Therefore, INFOBLEND theoretically reduces the computational overhead.
>
> ## Question 3
>
> > Figure 10 could have a more meaningful comparison if the value of k were similar for Cacheblend and INFOBLEND. I'd encourage the others to add this as currently the gains are not clearly visible and are hard to compare.
>
> The reason why we do not set similar value of $k$ for CacheBlend and INFOBLEND is that the generation quality of CacheBlend-1 is unacceptable, as shown in our response to Weakness 1. So it is meaningless to evaluate its throughput in Figure 10.
>
> We sincerely invite you to re-evaluate the paper. Please let us know if the response addresses the weakness.
>
> Best regards.

---

### Official Review · Reviewer_3iNp · 2025-10-30

**Soundness:** 2
**Presentation:** 1
**Contribution:** 1
**Rating:** 2
**Confidence:** 4

**Summary:**

This paper introduces INFOBLEND, a new system for caching and reusing key-value caches in multimodal large language model inference, especially for contexts with interleaved text and images. Unlike existing context caching methods that only reuse KV caches for exact prefix matches, INFOBLEND enables position-independent and partial reuse of KV caches, improving efficiency even when prompts differ slightly.

**Strengths:**

1. The MLLMs are promising.

2. The inference acceleration is interesting.

**Weaknesses:**

1. The writing quality is poor. Please revise the entire paper for improved logical flow. Also, it would be better to add a pipeline or an overview of the algorithms to clearly show how your model works.

2. The novelty of this paper is limited. The importance of the first token has been common knowledge for at least a year. Selecting partially important tokens for re-computation has also been investigated by existing papers, such as CacheLend. The novelty of this paper regarding related works is limited; it is mainly an extension of existing text LLM acceleration approaches to MLLMs.

3. The experiments are weak. Since this work focuses on LLM acceleration, it should be compared with GraphRAG and other related works that also accelerate LLM inference, as analyzed in the related work section.

4. While the paper discusses multimodal data, most experiments and analysis focus on images. The effectiveness for audio, video, or other modalities is not thoroughly evaluated.

5. Since this is a system-level optimization, you should compare your approach with VLLM. VLLM has been one of the most state-of-the-art baselines for such papers, even though it was published a year ago.

**Questions:**

See weakness

---

> ### Author Response · Authors · 2025-11-20
>
> We appreciate your valuable feedback. We will revise the paper to clarify the weaknesses.
>
> ## Weakness 1
>
> We will provide the following pseudocode of selective attention to clearly explain the algorithm. Moreover, Figure 7 in the paper shows the algorithm pipeline, and Figure 5 shows the overview of INFOBLEND.
>
> ```python
> """
> Input:
>     q (tensor): the recomputed q
>     k (tensor): the recomputed k
>     v (tensor): the recomputed v
>     kv_cache (`tuple` of tensor): the stored KV cache (already concatenated with dummy cache)
>     attn_metadata (`dict`): other parameters, including selected token ids.
> Output:
>     o (tensor): the output of attention algorithm
> """
> def attn_fwd_with_pic(q, k, v, kv_cache, attn_metadata):
>     key_cache, value_cache = kv_cache
>     # get the indices of important tokens, which are computed after tokenization
>     imp_indices = attn_metadata["imp_indicies"]
>     # replace the old KV cache with recomputed one
>     key_cache[imp_indices] = k
>     value_cache[imp_indices] = v
>     k = key_cache
>     v = value_cache
>
>     # attention algorithm
>     o = attn_fwd_func(q, k, v, attn_metadata)
>     return o
> ```
>
> ## Weakness 2
>
> Thank you for pointing out the similarity between CacheBlend and INFOBLEND. We will highlight the differences of multimodality here.
>
> 1. CacheBlend is not efficient enough from a system perspective. **The multimodal context is interleaved with mutable text, which is different from the scenario of RAG. CacheBlend assumes that all chunks are concatenated consecutively, without considering interleaved variable text.** The KV caches of images are stored, while those of text are not. Therefore, CacheBlend first computes the KV caches of text, and then reuses the KV caches of text and images. In order to prevent triggering MLLM engine twice, we design the selective attention as shown in Figure 7. The evaluation results illustrate its efficiency. We emphasize this issue of existing papers in Lines 92-95 and Lines 192-195.
>
> 2. In terms of selection strategy, CacheBlend merely focuses on numerical equivalence and tries to find the KV caches that lead to the largest deviation, without understanding what is behind these numerical deviations. We, however, pin down the root cause of this deviation (attention sink) and leverage this new cognitive to reduce redundant computation.
>
> 3. Last but not least, the computation overhead of INFOBLEND is theoretically less than that of CacheBlend. Suppose that the number of image tokens is $N$. Initially, computing the attention score for $N$ tokens has an $O(N^2)$ complexity.
> - INFOBLEND computes $k$ tokens for each image (Figure 7), leading to $O(kN)$ complexity.
> - CacheBlend computes $rN$ tokens for each image, leading to $O(rN^2)$ complexity.
>
> ## Weakness 3
>
> Following your recommendation, we investigate GraphRAG [1]. GraphRAG aims to answer global questions directed at an entire text corpus. It builds a graph index and pre-generate community summaries. Given a question, each community summary is used to generate a partial response, and the partial responses are combined and used to generate a final global answer.
>
> INFOBLEND focuses on reusing the KV cache of multimodal context. It is orthogonal to GraphRAG. For example, when GraphRAG uses partial responses to generate a final global answer, INFOBLEND can be applied to accelerate the generation.
>
> INFOBLEND is also compatible with many acceleration techniques. For example, INFOBLEND accelerates the prefill phase, while speculative decoding accelerates the decode phase. They can be applied together without conflict.
>
> [1] Edge, D., Trinh, H., Cheng, N., Bradley, J., Chao, A., Mody, A., Truitt, S., Metropolitansky, D., Ness, R. O., and Larson J. From Local to Global: A Graph RAG Approach to Query-Focused Summarization, 2025. URL https://arxiv.org/abs/2404.16130.
>
> ## Weakness 4
>
> We apologize for using the word "multimodal" ambiguously. We mistakenly follow many papers of the multimodality community that also claims to be multimodal but only considers texts and images. We will change the title and anything related to multimodal context caching. However, we argue that and image-text also belongs to multimodal situations. In our revisions, we will point out the specific modalities that we considered in this paper --- images and texts, explicitly.
>
> ## Weakness 5
>
> VLLM [2] is a state-of-the-art baseline, so we implement context caching (CC) algorithms based on VLLM 0.9.0 (mentioned in Line 395). As you know, prefix caching is enabled by default, so the results of the prefix caching is generated by vanilla VLLM.
>
> [2] Kwon, W., Li, Z., Zhuang, S., Sheng, Y., Zheng, L., Yu, C. H., Gonzalez, J. E., Zhang, H., and Stoica, I. Efficient memory management for large language model serving with pagedattention. In Proceedings of the ACM SIGOPS 29th Symposium on Operating Systems Principles, 2023.
>
> We will incorporate the clarifications and address the issues in the next draft. Thank you again.
>
> Best regards.

---

### Author Response · Authors · 2025-12-01
**Summary of the Paper and Discussions**

We thank all reviewers and AC for their efforts. To facilitate the AC's understanding and evaluation of our work, we summarize the paper and all reviews as follows.

## A key reason the paper merits acceptance

The paper proposes position-independent caching (PIC) for MLLMs, moving beyond the limitations of standard prefix caching. We are the first to study the PIC problem in the multi-modality field, to the best of our knowledge.

## Strengths

1. **Solid Background**: The MLLMs are promising and the KV cache size of images is growing. (Reviewer 3iNp & ozB4)

2. **Problem Significance**: Identifies the inefficiencies of existing method including prefix caching and prior works on RAG system in the field of multi-modality. (Reviewer ozB4 & a8xw & aAof)

3. **Novel Approach**: Proposes an interesting method (PIC) to reuse KV caches without positional restriction. (Reviewer 3iNp & aAof)

4. **Insightful Heuristic**: Finds out the root cause of KV deviation (attention sink) and leverages this new cognitive to reduce redundant computation. (Reviewer a8xw & aAof)

5. **Practical System Design**: Addresses system-level challenges using techniques such as parallel loading/computation and dummy caches. (Reviewer a8xw & aAof)

6. **Strong Empirical Results**: Demonstrates significant improvements in TTFT (up to 54.1% reduction) and throughput (up to 2.0x increase), while maintaining relatively high generation quality. (Reviewer a8xw & aAof)

## Weaknesses

We summarize the weaknesses and brief responses here. The detailed responses can be found following our references.

1. It is unclear how the partial reuse is applied. (Weakness 1 by 3iNp)

- **Brief response**: We have incorporated the pseudocode to facilitate understanding.

2. The differences from CacheBlend seem minor, and the challenges in the field of multi-modality are unclear. (Weakness 2 by 3iNp, Weakness 1 & 2 by a8xw)

- **Brief response**: The multimodal context is interleaved with mutable text, which is different from the scenario of RAG. CacheBlend assumes that all chunks are concatenated consecutively, without considering interleaved variable text. This approach is inefficient since it triggers the MLLM engine twice for prefilling. InfoBlend addresses this challenge through dummy caches.

3. Lacks comparison with GraphRAG and VLLM. (Weakness 3 & 5 by 3iNp)

- **Brief response**: GraphRAG is orthogonal to our work and VLLM has already been included in the baselines.

4. Lacks analysis of bigger models and other modalities. (Weakness 4 by 3iNp, Weakness 4 by a8xw, Weakness 2 by aAof)

- **Brief response**: The “attention sink” phenomenon emerges in bigger models and the LLM backbone of MLLM. InfoBlend leverages this insight and works.

5. The reason why the proposed selection strategy is more efficient is unclear. (Weakness 1 by ozB4, Weakness 4 by aAof)

- **Brief response**: CacheBlend merely focuses on numerical equivalence and tries to find the KV caches that lead to the largest deviation, without understanding what is behind these numerical deviations. We, however, pin down the root cause of this deviation (attention sink) and leverage this new cognitive to reduce redundant computation.

6. The scenarios where the accuracy-latency trade-off is acceptable should be discussed. (Weakness 2 by ozB4, Weakness 1 by aAof)

- **Brief response**: InfoBlend finds application in latency-sensitive scenarios such as game AI, voice assistant, and simultaneous interpretation. In these scenarios, users demand a rapid response and tolerate imperfect answers.

7. The different variants of InfoBlend seem similar. (Weakness 3 by a8xw)

- **Brief response**: The recomputation cost of InfoBlend is so low that even if we increase the number of tokens to recompute, the TTFT remains similar. We find that the answers of different InfoBlend variants are similar, so their scores on open questions are similar.

8. Open questions are evaluated based on GPT scores. (Weakness 3 by aAof)

- **Brief response**: GPT score is widely used by previous works when evaluating open questions.

We thank the reviewers for their valuable suggestions, which helped us improve our work. Due to the closure of the discussion phase, however, we were unable to engage in further communication. We would greatly appreciate it if the responses could earn the AC’s recognition.

Sincerely.

---

### Meta-Review · Area_Chair_VH34 · 2026-01-01

**Summary:**

This paper proposes InfoBlend, a system for position-independent KV cache reuse in multimodal large language model inference, with the goal of reducing prefill latency for interleaved text–image inputs. The paper addresses an important and timely systems problem, and several reviewers acknowledged the practical relevance of accelerating multimodal inference as well as the reported gains in latency and throughput. The system design is reasonably well thought out, and the authors provided detailed rebuttals clarifying algorithmic details and implementation choices.

**Reviewer Concerns:**

Despite the practical motivation, reviewers raised substantial concerns regarding the novelty, clarity, and generality of the contribution. A recurring issue is that the core idea was viewed as closely related to existing work such as CacheBlend, with the incremental differences not always clearly articulated or convincingly justified as multimodal-specific. Several reviewers found the writing and presentation difficult to follow, which further obscured the technical contributions. Concerns also remain about the strength of the experimental evaluation, including limited coverage of modalities beyond images, insufficient validation on larger or more diverse models, reliance on noisy GPT-based evaluation metrics, and a non-negligible accuracy degradation compared to prefix caching. While the rebuttal addressed many questions with additional explanations, these responses did not fully alleviate doubts about the paper’s conceptual novelty, empirical robustness, and overall maturity.

**Reviewer Scores:**

The reviewer scores were mixed but skewed toward rejection, with multiple reviewers rating the paper below the acceptance threshold and expressing high confidence in their assessments. While a small number of reviewers viewed the work more favorably and were open to acceptance, there was no strong consensus that the contribution clearly meets the standard for ICLR.

---

### Decision · Program_Chairs · 2026-01-26

Reject